# High Temperature Oxidation Behaviors of BaO/TiO_2_ Binary Oxide-Enhanced NiAl-Based Composites

**DOI:** 10.3390/ma14216510

**Published:** 2021-10-29

**Authors:** Bo Li, Ruipeng Gao, Hongjian Guo, Congmin Fan

**Affiliations:** 1State Key Laboratory for Mechanical Behaviour of Materials, School of Materials Science and Engineering, Xi’an Jiaotong University, Xi’an 710049, China; 2School of Mechanical and Precision Instrument Engineering, Xi’an University of Technology, Xi’an 710048, China; vessjessy@xaut.edu.cn; 3School of Bailie Mechanical Engineering, Lanzhou City University, Lanzhou 730070, China; guohj@lzcu.edu.cn; 4College of Materials and Chemistry & Chemical Engineering, Chengdu University of Technology, Chengdu 610059, China

**Keywords:** hot-press sintering, oxidation resistance performance, oxidation mechanism, element diffusion

## Abstract

High temperature lubricating composites have been widely used in aerospace and other high-tech industries. In the actual application process, high temperature oxidation resistance is a very importance parameter. In this paper, BaO/TiO_2_-enhanced NiAl-based composites were prepared by vacuum hot-press sintering. The oxidation resistance performance of the composites at 800 °C was investigated. The composites exhibited very good sintered compactness and only a few pores were present. Meanwhile, the composite had excellent oxidation resistance properties due to the formation of a dense Al_2_O_3_ layer which could prevent further oxidation of the internal substrate; its oxidation mechanism was mainly decided by the outward diffusion of Al and the inward diffusion of O. The addition of BaO/TiO_2_ introduced more boundaries and made the K_p_ value increase from 1.2 × 10^−14^ g^2^/cm^4^ s to 3.3 × 10^−14^ g^2^/cm^4^ s, leading to a slight reduction in the oxidation resistance performance of the composites—although it was still excellent.

## 1. Introduction

In order to meet the rapid development of aerospace, energy, chemical and other industries, high temperature structural materials with excellent comprehensive performance have been widely studied in recent years [1,2,3,4]. Numerous moving parts of these industrial fields, such as engineering, bearings, cutting tools, turbine blades, sealing devices, etc., are utilized in extremely harsh working environment [5,6,7,8]. Materials will experience severe high temperature friction and wear problems in these conditions. Therefore, it is very urgent to search for excellent high-temperature lubricating materials to meet the development needs of these industries. Composites consisting of a high temperature matrix and high temperature lubricant phase can be effectively designed to create promising high-temperature lubricating materials.

Lubrication at higher temperature can only be achieved by solid lubricants. Conventional solid lubricants including soft metals (Au, Ag, etc.), fluorides (BaF_2_, CaF_2_, etc.), graphite, MoS_2_ and certain metal oxides (V_2_O_5_, ZnO, etc.) have been designed at medium temperatures [8,9,10,11,12,13,14,15,16]. X. Shi et al. fabricated the M50–MoS_2_ self-lubricating composite, which possessed excellent tribological properties from 150 to 450 °C. However, some of their intrinsic deficiencies make it difficult to achieve effective lubrication at high temperatures. For example, Ag will migrate quickly at elevated temperatures [17]; graphite and MoS_2_ cannot present desirable lubrication at high temperatures because of poor oxidation resistance [8]. Recently, double metal oxides, which present good chemical and structural stability, strength and corrosion resistance, are easy to deform, shear and generate ternary oxide lubricating phases at high temperatures, and exhibit incredible tribological properties and oxidation resistance, have been widely added in high temperature matrix materials. Erdemir introduced the crystal chemistry model and found that the greater the potential difference between double metal oxides, the better the lubricating performance [18]. Stone et al. prepared a nano-silver-incorporating NbN nanocomposite film, which reacted above 700 °C to form a AgNbO_6_ lubricating metal oxide phase to keep its friction coefficient between 0.15–0.3 [17]. However, the relatively high migration rate of silver at high temperatures would destroy the surrounding components. Several works have found that Al_2_O_3_ and TiO_2_ react with each other to form Al_2_TiO_5_, and can improve high-temperature anti-wear and anti-friction properties [19,20]. X. Hua et al. fabricated nanostructured and conventional Al_2_O_3_-3 wt.% TiO_2_ coatings by atmosphere plasma spraying and investigated the effects of temperature on tribological properties [19]. However, the high melting point and poor high temperature softening of Al_2_O_3_ limit its mechanical processing performance [21]. In our recent work, BaO/TiO_2_ not only strengthened the NiAl matrix, but more importantly optimized its high-temperature friction properties [22]. In practical applications, these high temperature lubricating materials not only need to have good high temperature lubrication performance, but also need to have sufficiently high temperature oxidation resistance properties, which guarantee their working stability and service life. Nevertheless, these studies almost only focus on high temperature lubrication properties. Inevitably, these two properties must be met simultaneously in a high temperature environment, so the study of high temperature oxidation resistance of the composites should also be worthy of attention. It is very important to guide the application of the composites at high temperatures.

NiAl as a kind of intermetallic compound with a melting point of 1638 °C; it is a promising matrix material at elevated temperatures due to its notable mechanical strength and remarkably high temperature oxidation resistance [23,24,25,26,27,28,29]. Compared with other traditional Ni-based materials, the density of NiAl is about 2/3 of theirs, and so better in line with today’s lightweight and energy-saving trends [23,24].

In this paper, BaO/TiO_2_-enhanced NiAl-based composites were produced by vacuum hot-press sintering, and their structural features and oxidation behaviors were accordingly studied. At the same time, their high temperature oxidation mechanisms were analyzed.

## 2. Experimental Details

BaO/TiO_2_-enhanced NiAl-based composites were produced by vacuum hot-press sintering, with NiAl intermetallic, TiO_2_ and BaO powders as the raw powders. The molar ratio of BaO and TiO_2_ was 1:1. According to the different mass ratios of the enhanced phase and NiAl matrix, four different composites were designed in Table 1. Before sintering, ball milling was used to refine the NiAl intermetallic powders with a rotating speed of 250 rps/min for 10 h. The same ball milling process was used for subsequent mixing of the reinforcing phase and matrix powders. The mixed powders were enclosed and then sintered with sintering temperatures of 1300 °C, holding times of 1 h, pressures of 20 MPa, vacuum degrees of 10^−2^ Pa and cooling with a furnace.

The sample for the oxidation test obtained by vacuum hot-press sintering was processed into a strip sample of 20 mm × 10 mm × 2 mm by wire cut electrical discharge machining (WEDM), sanded by 180 mesh, 400 mesh, 600 mesh, 800 mesh, 1000 mesh sandpapers sequentially and polished to reduce surface roughness. Subsequently, these bulk samples were ultrasonic cleaned with absolute ethanol for 30 min. High temperature oxidation resistance tests were performed in a box-type resistance furnace, with oxidation temperatures of 800 °C, holding times from 20 h to 100 h, and intervals of 20 h.

The phase composition of these composites was characterized by an X-ray diffractometer (Philips X’Pert-MRD, Philips, Eindhoven, Netherlands) with a scan speed of 10°/min. The morphologies of the surfaces were observed by scanning electron microscopy (Zeiss Gemini SEM 500, Zeiss, Jena, Germany) with an energy dispersive Spectrometer (EDS, Zeiss, Jena, Germany) and the roughness of the surfaces was measured by a VK-9710 color 3D laser scanning confocal microscope (LSCM, Keyence, Osaka, Japan); each composite was measured 5 times and then averaged. The oxidation weight gain was measured by an electronic balance with an accuracy of 0.001 g.

## 3. Results and Discussion

### 3.1. Phases and Microscopic Morphologies of the Composites

Figure 1 shows the powders’ morphologies and element distribution after mixing, which presented with a uniform distribution and have average particle sizes. Besides this, according to the positional overlap of elements Ba and Ti, the TiO_2_ and BaO oxides combined very well. Figure 2 presents the XRD patterns of the composites after sintering. NiAl–BaO/TiO_2_ composites all contained strong NiAl peaks mainly. Meanwhile, except for the presence of intermetallic NiAl, there were also some peaks of BaO, TiO_2_ and BaTiO_3_ in NA1, NA2 and NA3. The existence of BaTiO_3_ peaks indicated that BaO and TiO_2_ reacted with each other at sintering temperatures of 1300 °C, and its intensity increased with the increased content of reinforcing phases. With the increasing of oxides content, NiAl peaks shifted to the left accordingly, which is probably due to the oxide elements dissolving in the NiAl matrix during sintering, making the interplanar spacing expand and then improving the mechanical properties [22].

Figure 3 presents the sintered microscopic morphologies of the composites. The NA had a dense microstructure (Figure 3a). The microscopic morphologies of the composites involved some white phases with the addition of BaO and TiO_2_, which were almost evenly distributed in the NiAl grain boundaries. Meanwhile, these white phases gradually increased with the increased content of BaO and TiO_2_. Although a few pores (black areas) appeared in the matrix materials, overall, the composite still exhibited a very good sintered compactness.

### 3.2. Oxidative Thermodynamics and Kinetics

The isothermal oxidation kinetics curves of the composites doped with different oxide contents are shown in Figure 4. Overall, the oxidation weight gains of all the samples at 800 °C are functions of oxidation time, and the samples of each component ratio follow a parabolic law, which can be described by the following oxidation rule formula [30]:(Δm/A)^2^ = K_p_t(1)
where Δm/A is weight gain per unit area; t is oxidation duration; and K_p_ is parabolic constant.

Table 2 calculates the K_p_ values of oxidized composites. The NA composite held the minimum K_p_ value of 1.2 × 10^−14^ g^2^/cm^4^ s, and the NA3 composite had the maximum K_p_ value of 3.3 × 10^−14^ g^2^/cm^4^ s. Increases in the content of the reinforcing phase made the value of K_p_ slightly increase, indicating that the oxidation behavior was intensified after the addition of BaO/TiO_2_—but the slow increase indicates that the effect was not obvious. In fact, the NA3 still had a good oxidation resistance performance.

Figure 5 presents XRD patterns of the composites after oxidation. Compared with XRD patterns after sintering, the main emerging crystal phase was Al_2_O_3_, and a small amount of NiO also existed. Besides this, NiAl_2_O_4_ peaks also appeared after oxidation. During the oxidation of the composites, the following reactions can easily occur [31,32,33,34]:2NiAl(s) + 3/2O_2_(g) → Al_2_O_3_(s) + 2Ni(s)(2)
Ni(s) + 1/2O_2_(s) → NiO(s)(3)
2Al(s) + 3/2O_2_(s) → Al_2_O_3_(s)(4)
Al_2_O_3_(s) + NiO(s) → NiAl_2_O_4_(s)(5)

Figure 6 shows the Gibbs free energy of different reactions at different temperatures [35]. It can be concluded that Al_2_O_3_ was extremely easy to form on the surface of the composites. The Gibbs free energy of reaction 5 was very high, meaning that this reaction was very difficult to carry out. As such, the amount of NiAl_2_O_4_ was supposed to be low. In summary, Al_2_O_3_ was the main oxidation product, which also is consistent with the XRD results.

In the initial stages of oxidation, the rate of oxidative weight gain of each sample was relatively fast. During this time period, Al_2_O_3_ gradually nucleated, growing at higher energy levels on the surface where exposed to air, especially at defects such as pores and phase boundaries. In these samples, because of good sintering compactness proven by the small number of pores (Figure 3), surface defects were mainly the effects of phase boundaries. These defects led to faster oxidation weight gain at early stages, which is consistent with the trend of morphologies after sintering. This process continued with the extension of the constant temperature holding time. Meanwhile, the growth of a nucleated aluminum oxide nucleus as described above began to become the main factor affecting oxidative weight gain, until the oxide film almost covered the whole surface as well as some interfaces. During this period, the concentration gradient of Ni caused by Al depletion would make Ni diffuse inward [30]. In the late stage of oxidation, the oxidation rate drops comparatively, due to Al outward diffusion and O inward diffusion determining the process rate [33,36]. Besides this, the diffusion of elements at the phase boundaries is faster than diffusion within the crystal, which also explains why the oxidation rate of NA3 is still much larger than others [33,36].

### 3.3. Cross-Sectional and Surface Morphologies of the Composites after Oxidation

To further investigate the oxidation performance of the composites, their cross-sectional morphologies were observed. Figure 7 shows cross-sectional morphologies of four composites after oxidation. After 100 h oxidation at 800 °C, the NA showed almost no obvious oxide layer, which shows that the oxidation resistance of NiAl was excellent. This result is also consistent with the oxidation weight gain curves of the composites (Figure 4). The thickness of the oxide film increased with the addition of BaO/TiO_2_. Meanwhile, increases in BaO/TiO_2_ led to the thickness of the oxide films to gradually increase. The NA3 had the thickest oxide film. It can be concluded that the NA had the best oxidation resistance and the NA3 had the worst oxidation resistance. These results are very consistent with the isothermal oxidation kinetics curves in Figure 4. The thickness of all the oxide films was less than 5 μm. It can be concluded that all composites had a good oxidation resistance performance at 800 °C.

Figure 8 displays the 3D laser scanning confocal microscope topography of the composites after oxidation for 100 h at 800 °C. The oxidation surface of NA contained a relatively compact and smooth oxide layer (Figure 8a). Nevertheless, the oxidation surface of the composites became rough with the addition of BaO/TiO_2_. The oxidation surface of the NA3 composite was the roughest (Figure 8d). Meanwhile, the roughness of oxidation surface of the composites gradually increased with the increasing content of BaO/TiO_2_. The roughness of the oxidation surface of the NA3 composite was the biggest, at 4.63 μm (Figure 8d). These results are consistent with the results of the oxidation weight gain curves of the composites (Figure 4).

## 4. Conclusions

In this paper, BaO/TiO_2_ enhanced NiAl-based composites were produced by vacuum hot-press sintering, and the structure features and oxidation behavior were studied. At the same time, their high temperature oxidation mechanisms were analyzed. The main conclusions are as follows:(1)The composites exhibited very good sintered compactness and only a few pores existed. With the increasing of oxide content, NiAl peaks shifted to the left accordingly, which can be explained by lattice distortion.(2)The composites had a good oxidation resistance at 800 °C, generating a dense Al_2_O_3_ film to prevent further oxidation. The oxidation products on the oxidation surface of the composites after oxidation for 100 h at 800 °C were mainly Al_2_O_3_, NiO and NiAl_2_O_4_.(3)The addition of BaO/TiO_2_ introduced more boundaries and made the K_p_ value increase from 1.2 × 10^−14^ g^2^/cm^4^ s to 3.3 × 10^−14^ g^2^/cm^4^ s, leading to a slight reduction of the oxidation resistance performance of the composite—although it was still excellent.(4)The thickness of the oxide film increased with the addition of BaO/TiO_2_. Meanwhile, increments of BaO/TiO_2_ led to the thickness of the oxide films to gradually increase and the oxidation resistance of the composites to gradually decrease.

## Figures and Tables

**Figure 1 materials-14-06510-f001:**
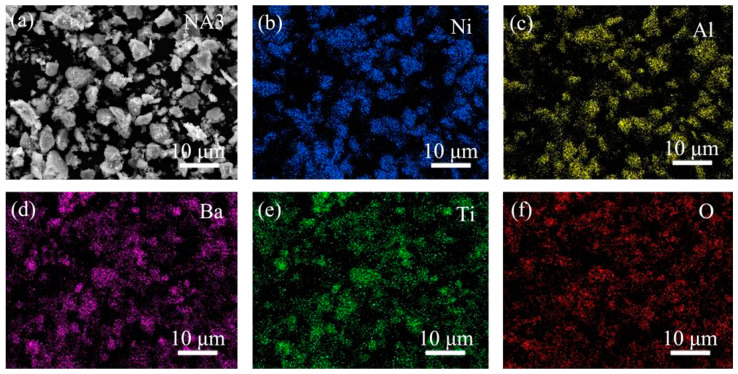
Morphology of mixed powders by ball milling: (**a**) SEM morphology; (**b**) Ni; (**c**) Al; (**d**) Ba; (**e**) Ti; (**f**) O.

**Figure 2 materials-14-06510-f002:**
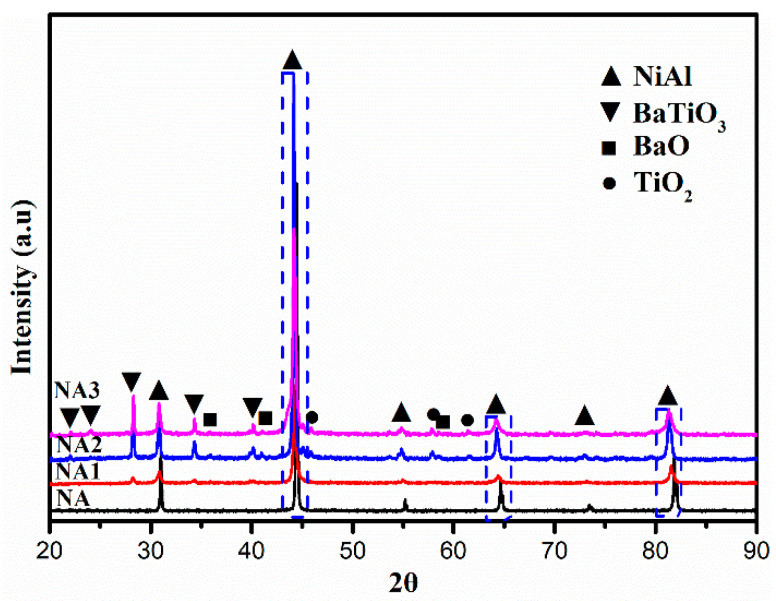
XRD patterns of the composites.

**Figure 3 materials-14-06510-f003:**
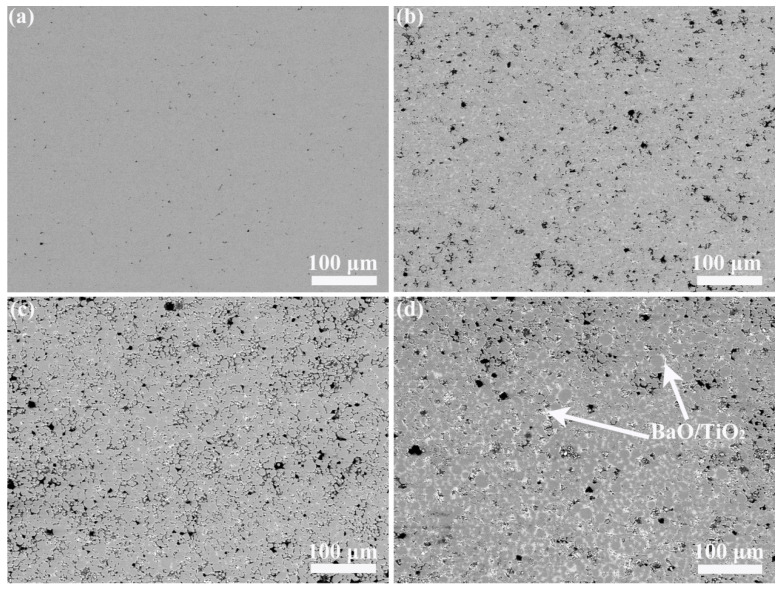
Sintering morphologies of the composites: (**a**) NA; (**b**) NA1; (**c**) NA2; (**d**) NA3.

**Figure 4 materials-14-06510-f004:**
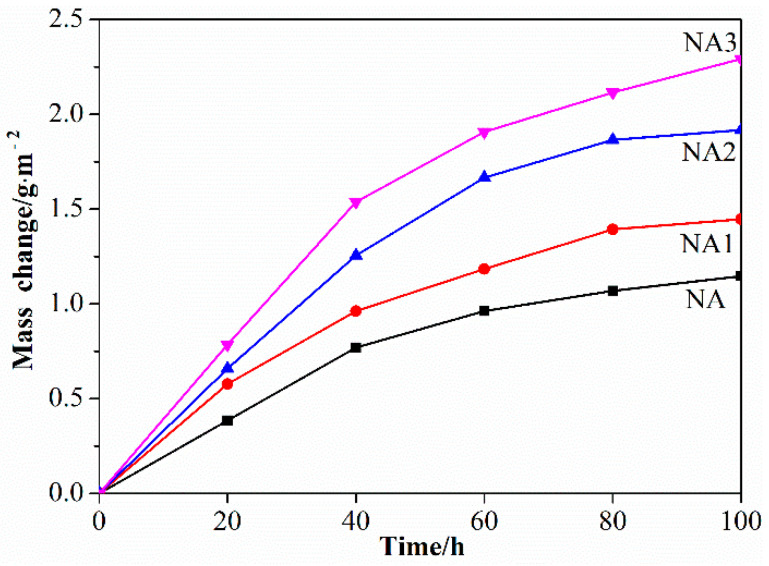
Oxidation kinetics curve of all composites.

**Figure 5 materials-14-06510-f005:**
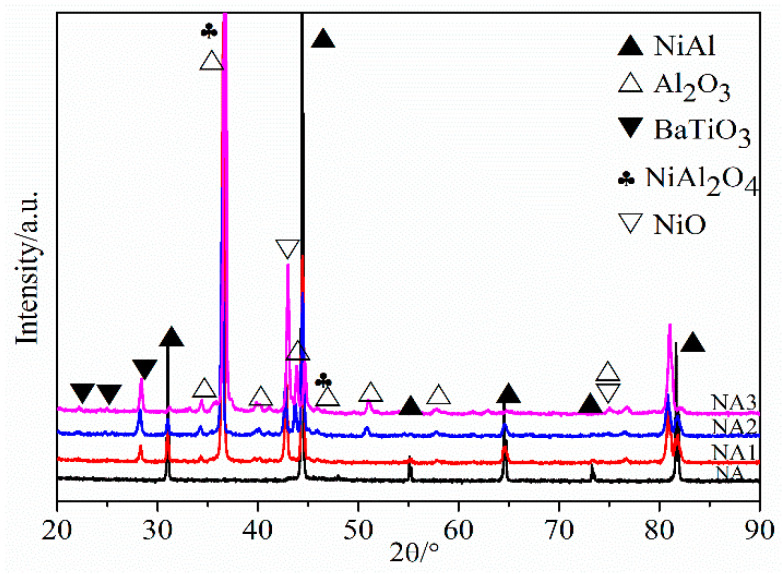
XRD patterns of the composites after oxidation.

**Figure 6 materials-14-06510-f006:**
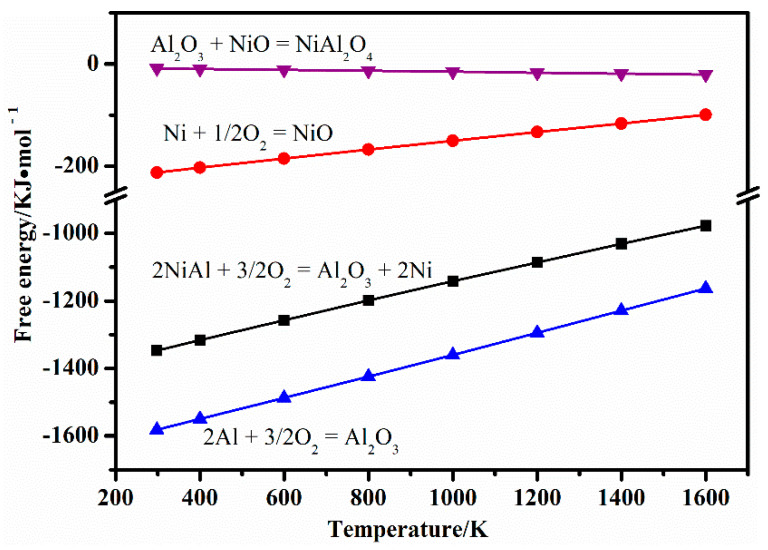
Gibbs free energy of different reactions at different temperatures [35].

**Figure 7 materials-14-06510-f007:**
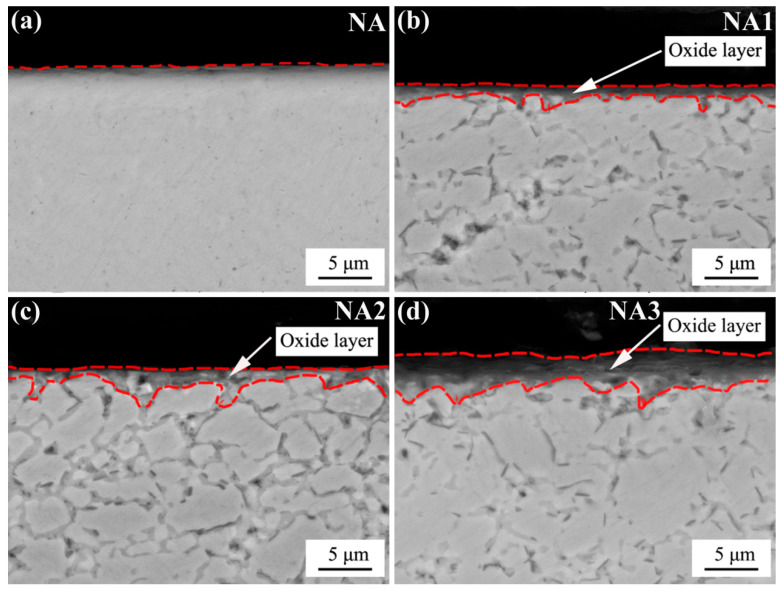
Oxidation cross-sectional morphologies of all composites: (**a**) NA; (**b**) NA1; (**c**) NA2; (**d**) NA3.

**Figure 8 materials-14-06510-f008:**
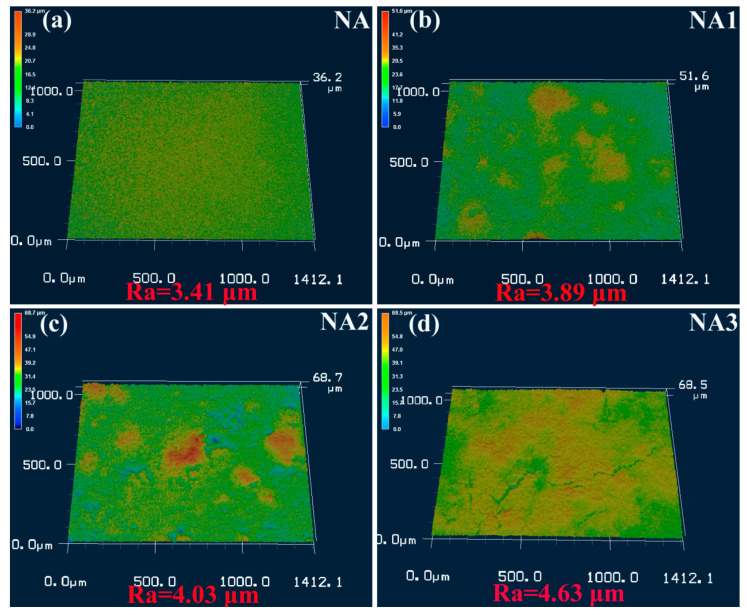
3D laser scanning confocal microscope topography of the composites after oxidation for 100 h at 800 °C: (**a**) NA; (**b**) NA1; (**c**) NA2; (**d**) NA3.

**Table 1 materials-14-06510-t001:** Composite composition.

Composites	NiAl (wt.%)	BaO (wt.%)	TiO_2_ (wt.%)
NA	100	0	0
NA1	90	6.6	3.4
NA2	80	13.2	6.8
NA3	70	19.8	10.2

**Table 2 materials-14-06510-t002:** Parabolic constants of all composites.

Composites	NA	NA1	NA2	NA3
K_p_/(g^2^/cm^4^ s)	1.2 × 10^−14^	2.6 × 10^−14^	3.1 × 10^−14^	3.3 × 10^−14^

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
