# Peer review of "High Temperature Oxidation Behaviors of BaO/TiO2 Binary Oxide-Enhanced NiAl-Based Composites"

_materials, 2021, doi:10.3390/ma14216510_

Round 1

Reviewer 1 Report

The manuscript presents the oxidation resistance and microstructural properties of NiAl-based intermetallic composites reinforced with BaO/TiO2 binary oxides. The results are new and convincing. The desired function could be exhibited finely, but some points should be added for readers' understanding prior to acceptance.

  • Title: it would be better to change to "....BaO/TiO2 binary oxides..." instead of "...double metal oxides of BaO/TiO2..."
  • Abstract: please mention the importance of the article briefly.
  • Introduction: please highlight the novelty of the work.
  • Results and discussion: page 3, line 108, it is worth mentioning that the lattice distortion and dislocation taken place in the composite structure are because of the parts of oxide element dissolve in the NiAl matrix during sintering.
  • Conclusion: I recommend adding a few sentences describing the main conclusions about the influence of powder composition and thickness of the oxide layer on the oxidation resistance.

Author Response

Q1: Title: it would be better to change to "....BaO/TiO2 binary oxides..." instead of "...double metal oxides of BaO/TiO2..."

   Response: According to the reviewer’s suggestion, we have changed the title of this paper.

Q2: Abstract: please mention the importance of the article briefly.

Response: The reviewer’s comments are of scientifically importance. According to the reviewer’s suggestion, we have added the importance of the article briefly in the abstract in the revised manuscript.

High temperature lubricating composites were widely used in aerospace and other high-tech industries. In the actual application process, high temperature oxidation resistance is a very importance performance.

Q3: Introduction: please highlight the novelty of the work.

Response: According to the reviewer’s suggestion, we have added the novelty of the work in the revised manuscript.

Q4: Results and discussion: page 3, line 108, it is worth mentioning that the lattice distortion and dislocation taken place in the composite structure are because of the parts of oxide element dissolve in the NiAl matrix during sintering.

   Response: According to the reviewer’s suggestion, we have revised this sentence in the revised manuscript.

Q5: Conclusion: I recommend adding a few sentences describing the main conclusions about the influence of powder composition and thickness of the oxide layer on the oxidation resistance.

Response: According to the reviewer’s suggestion, we have added a few sentences describing the main conclusions about the influence of powder composition and thickness of the oxide layer on the oxidation resistance in the revised manuscript.

The thickness of the oxide film increase with addition of BaO/TiO2. Meanwhile, increment of BaO/TiO2 leads to the thickness of the oxide films gradually increasing and the oxidation resistance of the composites gradually decreasing.

Reviewer 2 Report

The manuscript can be reconsidered after a major revision.

  1. The introduction part should be improved by reviewing more related works.
  2. The display resolution of all figures should be significantly improved.
  3. What is the time at which Kp values in Table 2 are evaluated? Is Kp a constant value?
  4. There is a replication of equation number.

Author Response

Q1: The introduction part should be improved by reviewing more related works.

Response: According to the reviewer’s suggestion, we have added more related works in the revised manuscript.

Q2: The display resolution of all figures should be significantly improved.

Response: According to the reviewer’s suggestion, we have improved the display resolution of all figures in the revised manuscript.

Q3: What is the time at which Kp values in Table 2 are evaluated? Is Kp a constant value?

Response: Actually, (Dm/A)2=Kpt , Dm is weight gain after oxidation test, A is oxidation surface area of the sample, t is oxidation duration. The Kp values in Table 2 are evaluated after oxidation test. Kp is not a constant value.

Q4: There is a replication of equation number.

Response: We are very sorry for our careless. We have revised it in the revised manuscript.

Reviewer 3 Report

The paper discusses oxidation behavior of BaO/TiO2 enhanced NiAl-based composites at high temperature. Four compositions of different ratios of BaO/TiO2 and NiAl were considered. Phases and microscopic morphologies of the composites powders were studied. Oxidation behavior was intensified due to addition of BaO/TiO2 but all composites show good oxidation resistance. Oxidation surfaces of the composites become rough with this  addition however the thickest oxide layer was less than 5 micrometers.

Possible applications of discussed composites as a solid lubricant in heavy duty tribo-fatigue systems could be referred to in the paper:

 (i) a method of experimental study of friction in a active system, (ii) state of volumetric damage of tribo-fatigue system, (iii)  spatial stress-strain state of tribofatigue system in roll-shaft contact zone, (iv) modeling of the damaged state by the finite-element method on simultaneous action of contact and noncontact loads, (v) tribo-fatigue behavior of austempered ductile iron monica as new structural material for rail-wheel system, (vi) research on tensile behaviour of new structural material monica, (vii) measurement and real time analysis of local damage in wear-and-fatigue tests.

The paper “High temperature oxidation behaviors of double metal oxides of BaO/TiO2 enhanced NiAl-based composites” could be published in Materials after minor revision.

Author Response

The paper discusses oxidation behavior of BaO/TiO2 enhanced NiAl-based composites at high temperature. Four compositions of different ratios of BaO/TiO2 and NiAl were considered. Phases and microscopic morphologies of the composites powders were studied. Oxidation behavior was intensified due to addition of BaO/TiO2 but all composites show good oxidation resistance. Oxidation surfaces of the composites become rough with this  addition however the thickest oxide layer was less than 5 micrometers.

Possible applications of discussed composites as a solid lubricant in heavy duty tribo-fatigue systems could be referred to in the paper:

(i) a method of experimental study of friction in a active system, (ii) state of volumetric damage of tribo-fatigue system, (iii)  spatial stress-strain state of tribofatigue system in roll-shaft contact zone, (iv) modeling of the damaged state by the finite-element method on simultaneous action of contact and noncontact loads, (v) tribo-fatigue behavior of austempered ductile iron monica as new structural material for rail-wheel system, (vi) research on tensile behaviour of new structural material monica, (vii) measurement and real time analysis of local damage in wear-and-fatigue tests.

The paper “High temperature oxidation behaviors of double metal oxides of BaO/TiO2 enhanced NiAl-based composites” could be published in Materials after minor revision.

Response: Thank you very much. According to the reviewer’s suggestion, we have revised this paper very carefully. Revised portion are marked in RED in the revised manuscript.

Reviewer 4 Report

The manuscript “High temperature oxidation behaviors of double metal oxides 2 of BaO/TiO2 enhanced NiAl-based composites” is prepared with less care. No novelty was found and authors repeated their own data from their previous article. Authors claimed that “In this paper, BaO/TiO2 enhanced NiAl-based composites were fabricated by vacuum hot-pressing sintering, and its structure features and oxidation behaviors were accordingly investigated.”  BaO/TiO2 –NiAl composite is published before by one of the authors. Figure 1 and xrd of the same material almost explained in published article. Mapping color can be changed and no relevance, instead you can give the direct spectrum. This manuscript needs more original data to be published in this journal. So I recommend rejection.

Author Response

The manuscript “High temperature oxidation behaviors of double metal oxides 2 of BaO/TiO2 enhanced NiAl-based composites” is prepared with less care. No novelty was found and authors repeated their own data from their previous article. Authors claimed that “In this paper, BaO/TiO2 enhanced NiAl-based composites were fabricated by vacuum hot-pressing sintering, and its structure features and oxidation behaviors were accordingly investigated.”  BaO/TiO2 –NiAl composite is published before by one of the authors. Figure 1 and xrd of the same material almost explained in published article. Mapping color can be changed and no relevance, instead you can give the direct spectrum. This manuscript needs more original data to be published in this journal. So I recommend rejection.

Response: In our previous article, we mainly investigated the microstructure, mechanical and tribological properties of this composites. However, we mainly researched the oxidation behavior of this composites at high temperature in this paper. It’s completely different from previous studies. 

Reviewer 5 Report

Congratulation for your work!

Round 2

Reviewer 2 Report

The revised manuscript can be accepted for publication.

Reviewer 4 Report

For the manuscript “High temperature oxidation behaviors of double metal oxides 2 of BaO/TiO2 enhanced NiAl-based composites”, still need to be improved a lot. In the introduction and experimental they claimed " BaO/TiO2 enhanced NiAl-based composites were produced by vacuum hot-pressing sintering.." with out citing their own published article which will make an impression that its a part of a novelty in this article.  Figure 1 is not convincing and need strong characterization on the composites. Why authors didnot give the EDX spectrum atleast instead of images. Authors failed to answer the reviewer's comments well. The manuscript  cannot be published in Materials as in the present form.